OBSERVATION

# Reverted HIV-1 Mutants in CD4+ T-Cells Reveal Critical Residues in the Polar Region of Viral Envelope Glycoprotein

Wuxun Lu,[a] Tai-Wei Li,[a] Stacia Phillips,[b] Li Wu[b]

[a]Center for Retrovirus Research, Department of Veterinary Biosciences, The Ohio State University, Columbus, Ohio, USA
[b]Department of Microbiology and Immunology, Carver College of Medicine, The University of Iowa, Iowa City, Iowa, USA

Wuxun Lu and Tai-Wei Li contributed equally to this article. Author order was determined in order of increasing seniority.

**ABSTRACT** HIV-1 envelope glycoprotein (Env) interacts with cell surface receptors and induces membrane fusion to enter cells and initiate infection. HIV-1 Env on virions comprises trimers of the gp120 and gp41 subunits. The polar region (PR) in the N-terminus of gp41 is composed of 17 conserved residues, including seven polar amino acids. We have reported that the PR is crucial for Env trimer stability and fusogenicity. Mutations of three highly conserved residues (S534P, T536A, or T538A) in the PR of HIV-1$_{NL4-3}$ significantly decrease or eliminate viral infectivity due to defective fusion and increased gp120 shedding. To identify compensatory Env mutations that restore viral infectivity, we infected a CD4+ T-cell line with PR mutants pseudotyped with wild-type (WT) HIV-1 Env or vesicular stomatitis virus envelope glycoprotein (VSV-G). We found that PR mutant-infected CD4+ T-cells produced infectious viruses at 7 days postinfection (dpi). Sequencing of the *env* cDNA from cells infected with the recovered HIV-1 revealed that the S534P mutant reverted to serine or threonine at residue 534. Interestingly, the combined PR-mutant HIV-1 (S534P/T536A or S534P/T536A/T538A) recovered its infectivity and reverted to S534, but maintained the T536A or T538A mutation, suggesting that HIV-1 replication in CD4+ T-cells can tolerate T536A and T538A Env mutations, but not S534P. Moreover, VSV-G-pseudotyped HIV-1 mutants with a fusion-defective Env also recovered infectivity in CD4+ T-cells through reverted Env mutations. These new observations help define the Env residues critical for HIV-1 infection and demonstrate that Env-defective HIV-1 mutants can rapidly regain replication competency in CD4+ T-cells.

**IMPORTANCE** Our previous mutagenesis study revealed that serine at position 534 of HIV-1 Env is critical for viral infectivity. We found that HIV-1 Env containing serine to proline mutation at position 534 (S534P) are incapable of supporting virus-cell and cell-cell fusion. To investigate whether these mutant viruses can recover infectivity and what amino acid changes account for recovered infectivity, we infected CD4+ T-cells with Env-mutant HIV-1 pseudotyped with WT HIV-1 Env or VSV-G and monitored cultures for the production of infectious viruses. Our results showed that most of the pseudotyped viruses recovered their infectivity within 1-week postinfection, and all the recovered viruses mutated proline at position 534. These observations help define the Env residues critical for HIV-1 replication. Because Env-defective HIV-1 mutants can rapidly regain replication competency in CD4+ T-cells, it is important to carefully monitor viral mutations for biosafety consideration when using HIV-1-derived lentivirus vectors pseudotyped with Env.

**KEYWORDS** HIV-1, envelope glycoprotein, polar region, pseudotyping, reversion mutations, replication, kinetics.

Address correspondence to Li Wu, li-wu@uiowa.edu.

The authors declare no conflict of interest.

HIV-1 envelope glycoprotein (Env) comprises a surface unit (gp120) and a transmembrane unit (gp41). HIV-1 Env forms trimers incorporated into virions to mediate receptor recognition and fusion-based entry (1–3). The gp41 contains an N-terminal ectodomain, a

transmembrane domain, and a C-terminal cytoplasmic tail. The ectodomain of gp41 is comprised of the N-terminal fusion peptide, polar region (PR), N-terminal heptad repeat, C-terminal heptad repeat, and membrane proximal external region (4). The PR has 17 residues including 7 polar amino acids (5, 6). Previous studies suggested that the PR participates in Env peptide-based membrane fusion (5–9); however, the function of the PR is not fully understood. We have reported that the PR is highly conserved and determines viral fusion and infectivity (10). We characterized three PR mutants (10) containing S534P/T536A, S534P/T536A/T538A or S534P, named M1, M3 or M4, respectively (Fig. 1A). These mutant Env proteins are unable to mediate virus–cell or cell–cell fusion, whereas mutant viruses pseudotyped with wild-type (WT) Env from HIV-1$_{NL4-3}$ can efficiently infect target cells (10). To examine whether fusion-defective PR mutant HIV-1 can regain infectivity through reversion mutations, we infected a human CD4$^+$ T-cell line with HIV-1 PR mutants pseudotyped with WT HIV-1 Env or vesicular stomatitis virus envelope glycoprotein (VSV-G).

We first generated single-cycle HIV-1 pseudotyped with WT Env from HIV-1$_{NL4-3}$ (termed M1/Env, M3/Env, M4/Env) as described (10) and infected CD4$^+$ Hut/CCR5 T-cells (11) to allow for potential reversion mutations that rescue viral infectivity. We then determined the infectivity of WT HIV-1 and mutants by infecting TZM-bl cells that express luciferase upon productive HIV-1 infection (12) (Fig. 1B). WT HIV-1 infection of Hut/CCR5 cells induced severe cytopathic effects at 3 days postinfection (dpi) as indicated by syncytia (Fig. 1C, indicated by arrows and zoom-in images). Hut/CCR5 cells infected with M1/Env, M3/Env, M4/Env mutants developed syncytia at 7 dpi (Fig. 1C), suggesting production of fusion-competent viruses and the slow occurrence of reversion mutations of Env.

To measure the infectivity of HIV-1 released by Hut/CCR5 cells, supernatant viral p24 (capsid) levels were quantified and equal amounts of WT or mutant HIV-1 were used to infect TZM-bl cells that express luciferase upon productive HIV-1 infection (12). Supernatants of HIV-1-infected Hut/CCR5 cells were collected every 2 days from 3 to 15 dpi to quantify viral infectivity using TZM-bl cells with equal p24 input. Overall, M1/Env, M3/Env and M4/Env showed slower kinetics of infectivity compared to WT HIV-1 (Fig. 1D). At 13 dpi, M3/Env showed highest infectivity and reached the level of infection equivalent to that of WT HIV-1 at 3 dpi, while M1/Env and M4/Env showed approximately 50% infectivity of M3/Env. These results suggest that it takes approximately 13 days for these fusion-defective HIV-1 mutants to regain infectivity in Hut/CCR5 cells.

To identify sequence changes in Env of the reversion mutants, we cloned full-length *env* genes of the recovered viruses and sequenced 800–900 nt covering the PR. To eliminate noninfectious virus particles produced by Hut/CCR5 cells, the collected M1/Env, M3/Env, and M4/Env viruses were used to infect fresh Hut/CCR5 cells. Infected Hut/CCR5 cells were collected at 3 dpi for cellular DNA extraction, and full-length *env* genes were cloned from integrated proviral DNA. For each recovered virus, six clones of the *env* gene were sequenced. Interestingly, the PR sequences of recovered viruses are identical to that of WT Env (Fig. 1E). Moreover, cloned *env* genes from the reversion mutants were also identical to WT *env* in the sequenced region outside of PR. Because the original mutants (M1, M3, and M4) and WT Env are identical aside from the PR point mutations sites (10), it is possible that HIV-1 recombination occurred between WT and mutant Env in the virus producer HEK293T cells during transfection, and recombinant replication-competent HIV-1 were then expanded in infected Hut/CCR5 cells. It is likely that, once an infectious HIV-1 was present in cultures, it started replicating efficiently in Hut/CCR5 cells.

Next, to exclude the possibility of recombination between mutant and WT Env, VSV-G-pseudotyped mutant viruses were generated in HEK293T cells by transfection, and then used to infect Hut/CCR5 cells to determine replication kinetics and *env* sequence changes (Fig. 2A). These VSV-G-pseudotyped mutant viruses were termed M1/VSV-G, M3/VSV-G, and M4/VSV-G. Hut/CCR5 cells were infected and supernatants were collected at 3 to 15 dpi to measure kinetics of HIV-1 replication using TZM-bl cells (Fig. 2A). At 3 dpi, WT HIV-1 infection induced syncytia formation in Hut/CCR5 cells as a

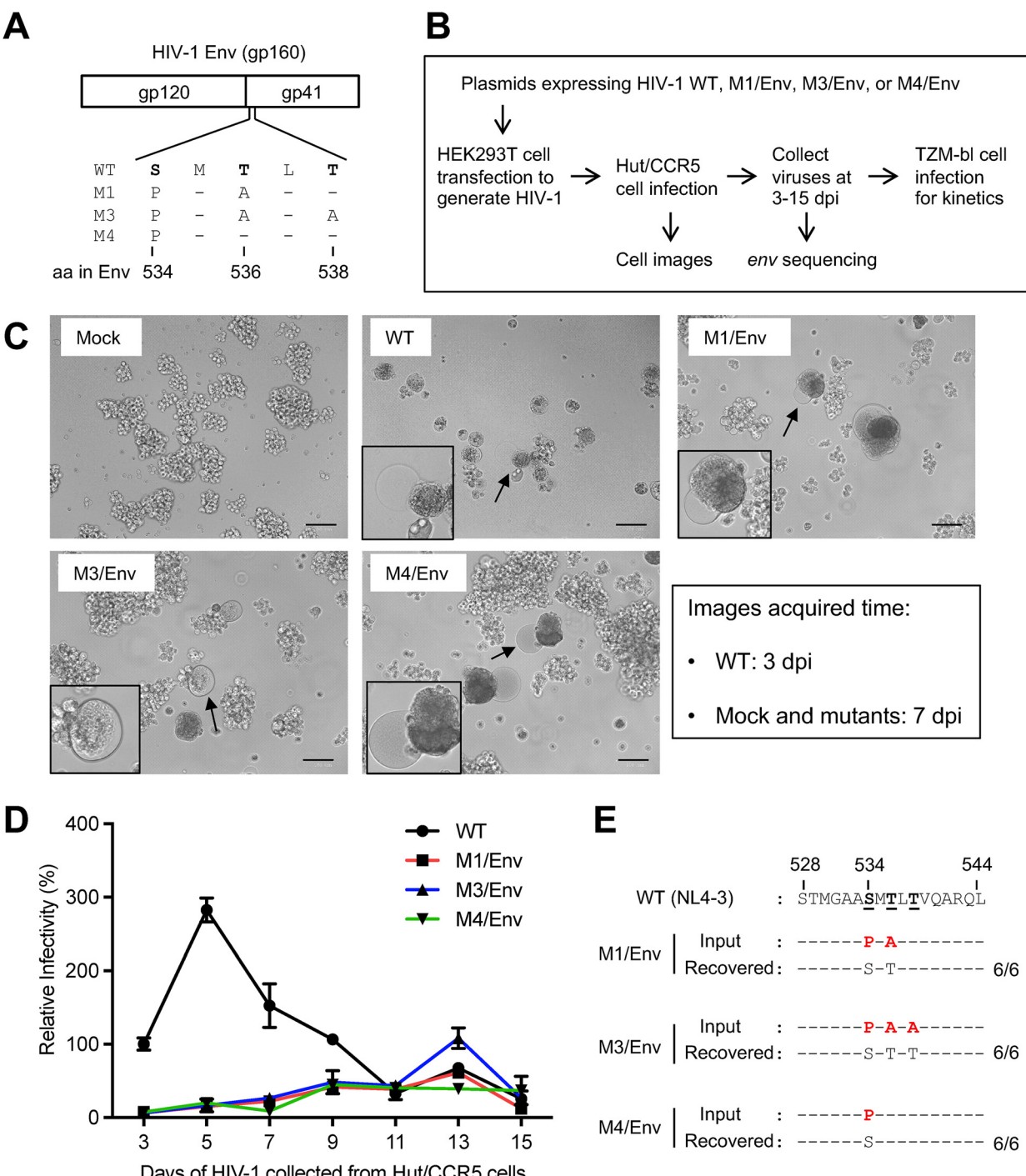

**FIG 1** Fusion-defective mutant HIV-1 pseudotyped with wild-type Env recover infectivity in Hut/CCR5 cells. (A) Amino acid (aa) changes in the PR of fusion-defective HIV-1 mutants (M1, M3, or M4) compared to wild-type (WT) HIV-1$_{NL4-3}$ (GenBank accession number M19921.2). The standardized aa numbers of Env are used (15), which are two numbers greater than those of the Env of HIV-1$_{NL4-3}$ (10). (B) Timeline and procedure of HIV-1 infection assays. (C) Morphology of Hut/CCR5 cells at 7 dpi (mock and mutant HIV-1 infections) or 3 dpi (WT HIV-1 infection). Cells with significant cytopathic effects are indicated by arrows and zoom-in images are shown in the insets. Scale bar: 100 $\mu$m. (D) Kinetics of WT and mutant virus infectivity during passaging in Hut/CCR5 cells. M1, M3, or M4 pseudotyped with WT Env were used to infect Hut/CCR5 cells with equal amounts of viruses (400 ng p24). WT HIV-1$_{NL4-3}$ was used as a positive control in the infection assay. Supernatants of HIV-1-infected Hut/CCR5 cells were collected every 2 days from 3 to 15 dpi to quantify viral infectivity using TZM-bl cells with equal viral input (0.2 ng p24). The infectivity of WT HIV-1 at 3 dpi was set as 100% and the relative infectivity is shown. All experiments were performed with triplicate samples and repeated two times, and means ± standard errors are shown. (E) Sequence alignment of the PR of recovered replication-competent HIV-1. Six clones were randomly selected from each mutant HIV-1 for sequencing (800-900 nt in *env* covering the PR). The mutated residues compared to WT HIV-1 sequence are shown in red.

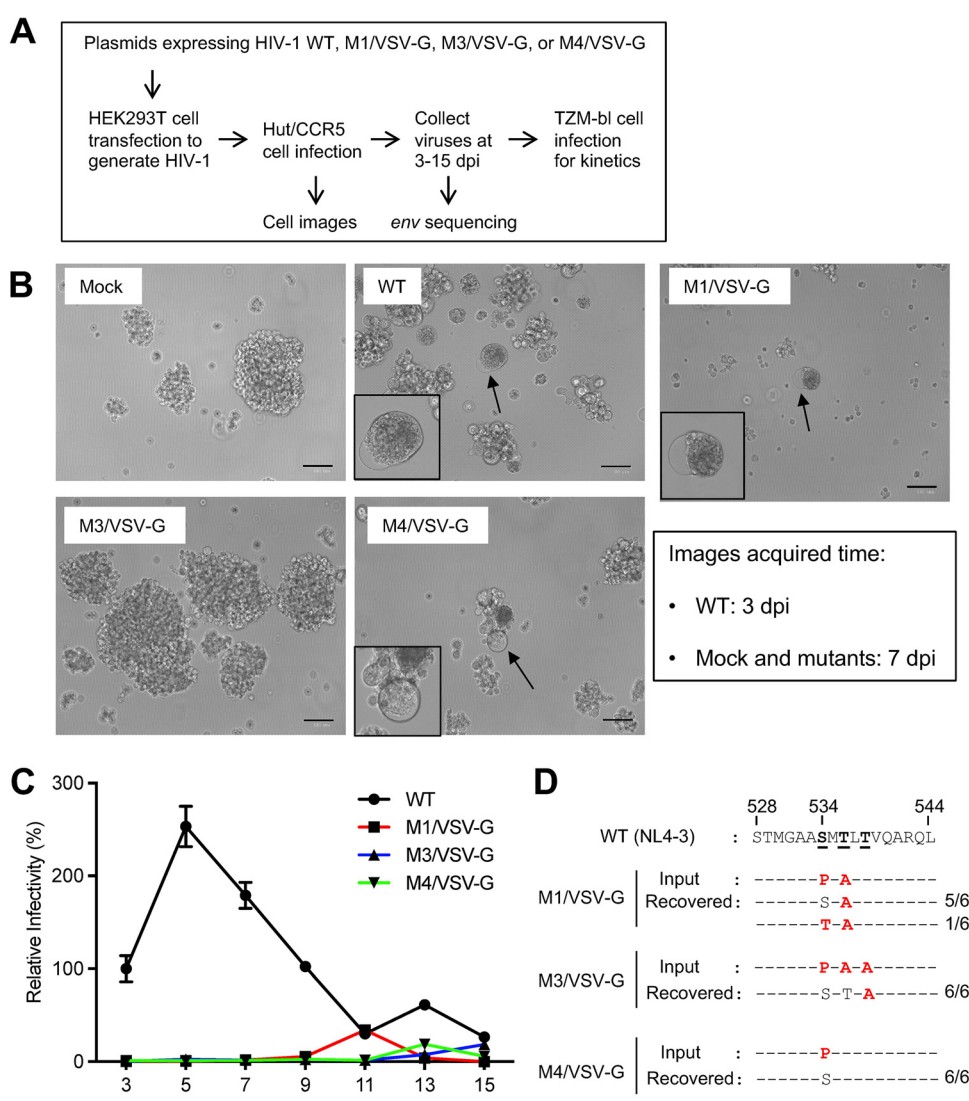

FIG 2 Fusion-defective mutant HIV-1 pseudotyped with VSV-G recover infectivity in Hut/CCR5 cells. (A) Timeline and procedure of HIV-1 infection assays. (B) Morphology of Hut/CCR5 cells at 3 dpi (WT infection) or 7 dpi (mock and mutant infections). Cells with significant cytopathic effects are indicated by arrows and zoom-in images are shown in the insets. Scale bar: 100 $\mu$m. (C) M1, M3, or M4 pseudotyped with VSV-G were used to infect Hut/CCR5 cells with equal amounts of viruses (400 ng p24). WT HIV-1$_{NL4-3}$ was used as a positive control in the infection assay. Supernatants of HIV-1-infected Hut/CCR5 cells were collected every 2 days from 3 to 15 dpi to quantify viral infectivity using TZM-bl cells with equal viral input (0.2 ng p24). The infectivity of WT HIV-1 at 3 dpi was set as 100% and the relative infectivity is shown. All experiments were performed with triplicate samples and repeated two times, and means $\pm$ standard errors are shown. (D) Sequence alignment of the PR of recovered replication-competent HIV-1. Six clones were randomly selected from each mutant for sequencing. The standardized aa numbers of Env are used (15), which are two numbers greater than those of the Env of HIV-1$_{NL4-3}$ (10). The mutated residues compared to WT HIV-1 sequence are shown in red.

positive control. At 7 dpi, large syncytia and significant cell death were observed in cells infected with M1/VSV-G or M4/VSV-G, but not with M3/VSV-G (Fig. 2B). However, syncytia also appeared in M3/VSV-G-infected Hut/CCR5 cells at 10 dpi (images not shown).

To determine the infectivity of VSV-G-pseudotyped viruses in Hut/CCR5 cells from 3 to 15 dpi, supernatants of Hut/CCR5 cells were collected every 2 days to quantify viral infectivity using TZM-bl cells with equal viral p24 input. Consistent with Env-pseudo-typed HIV-1 infection of Hut/CCR5 cells (Fig. 1D), M1/VSV-G, M3/VSV-G, and M4/VSV-G showed slower kinetics of infection compared with WT HIV-1 (Fig. 2C). M1/VSV-G showed similar infectivity comparable with WT HIV-1 at 11 dpi, suggesting that

reversion leading to the production of infectious virus does not occur until after 9 dpi. Interestingly, VSV-G-pseudotyped mutants showed different kinetics of recovered infectivity, with M1/VSV-G reaching peak infectivity at 11 dpi, followed by M4/VSV-G at 13 dpi, and M3/VSV-G at 15 dpi (Fig. 2C). Furthermore, pseudotyping PR mutant viruses with HIV-1 Env led to the appearance of infectious viruses earlier than VSV-G-pseudotyped mutant viruses (Fig. 1D and 2C). This difference might result from potential *env* gene recombination in virus producer HEK293T cells when generating Env-, but not VSV-G-pseudotyped HIV-1.

To identify sequence changes in the *env* gene of replication-competent HIV-1 recovered from M1/VSV-G, M3/VSV-G and M4/VSV-G, fresh Hut/CCR5 cells were infected with the recovered viruses. Cellular DNA was extracted at 3 dpi for cloning of full-length *env* genes from integrated proviral DNA. For each recovered virus, 6 clones of the *env* gene were sequenced. All of the recovered viruses mutated proline at position 534 (Fig. 2D). All but one (17/18, ~94.4%) sequenced clone indicated reversion of the proline at position 534 to the WT serine, confirming that original S534 is important for Env function and HIV-1 infectivity. One clone (1/18, ~5.6%) derived from M1/VSV-G infection revealed a P534T mutation (Fig. 2D). These results are in agreement with our previously published analysis (10) showing that the frequency of S534 and T534 in the PR of 57,645 HIV-1 Env sequences is 96.6% and 0.03%, respectively. Of note, T536A mutant remained the same in all 6 clones derived from M1/VSV-G but reverted to WT T536 in all 6 clones derived from M3/VSV-G infection (Fig. 2D). Our previous analysis showed that T536A variant also exists with a low frequency in 57,645 HIV-1 Env sequences from the Los Alamos National Laboratory HIV Sequence Database (10).

Our structural analysis (10) showed that the PR interacts with the C-terminal heptad repeat element of a neighboring protomer in gp41 to stabilize the gp120-gp41 trimer. These three PR mutations can alter the helical structure of the PR and destabilize the Env trimer and affect gp120-gp41 association (10). Thus, the PR of gp41, particularly the key residue S534, is structurally essential for maintaining HIV-1 Env trimer, viral fusogenicity and infectivity (10).

In summary, our new observations further define the Env residues critical for HIV-1 infection and demonstrate that Env-defective HIV-1 mutants can rapidly regain replication competency in CD4$^+$ T-cells. Our results also highlight the importance of carefully monitoring viral mutations for biosafety consideration when using HIV-1-derived lentivirus vectors pseudotyped with Env (13, 14).

## SUPPLEMENTAL MATERIAL

Supplemental material is available online only.
**SUPPLEMENTAL FILE 1**, PDF file, 0.1 MB.

## ACKNOWLEDGMENTS

We thank Eric Freed and Vineet KewalRamani for reagents, and members of the Wu lab for helpful discussions. This work was supported by NIH grants AI150343 and AI141495 (L.W.). The TZM-bl cells were obtained through the NIH AIDS Reagent Program, Division of AIDS, NIAID, NIH, from John C. Kappes, Xiaoyun Wu and Tranzyme Inc. W.L. and T.L. performed all experiments. W.L. and S.P. drafted the manuscript. L.W. conceived the study, supervised the work, and wrote the paper. All authors contributed to manuscript editing and revision.

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
