## [Reviewer comments · Microbiology Spectrum]

Microbiology Spectrum

Reverted HIV-1 mutants in CD4⁺ T-cells reveal critical residues in the polar region of viral envelope glycoprotein

Li Wu, Wuxun Lu, Tai-Wei Li, and Stacia Phillips

Corresponding Author(s): Li Wu, The University of Iowa

Review Timeline:

Submission Date:	September 22, 2021
Editorial Decision:	November 1, 2021
Revision Received:	November 5, 2021
Accepted:	November 16, 2021

Editor: Yongjun Sui

Reviewer(s): The reviewers have opted to remain anonymous.

Transaction Report:

DOI: <https://doi.org/10.1128/spectrum.01653-21>

November 1, 2021

Dr. Li Wu
The University of Iowa
Microbiology and Immunology
51 Newton Road
Iowa City 52242

Re: Spectrum01653-21 (Reverted HIV-1 mutants in CD4⁺ T-cells reveal critical residues in the polar region of viral envelope glycoprotein)

Dear Dr. Li Wu:

Thank you for submitting your manuscript to Microbiology Spectrum. When submitting the revised version of your paper, please provide (1) point-by-point responses to the issues raised by the reviewers as file type "Response to Reviewers," not in your cover letter, and (2) a PDF file that indicates the changes from the original submission (by highlighting or underlining the changes) as file type "Marked Up Manuscript - For Review Only". Please use this link to submit your revised manuscript - we strongly recommend that you submit your paper within the next 60 days or reach out to me. Detailed information on submitting your revised paper are below.

Link Not Available

Sincerely,

Yongjun Sui

Journals Department
Reviewer comments:

Reviewer #1 (Comments for the Author):

The paper presented by Wu and colleagues is a follow-up to a study previously conducted in the lab, which showed the critical role of the Env polar region (PR) of HIV-1 in controlling fusion between virus and cells and between cells. They previously have identified three mutants that they named M1, M3, and M4, all completely unable to fuse with target cells. However, these mutants, if pseudotyped with the HIV-1 WT Env protein, were fusogenic.

To examine whether M1, M3, and M4 PR mutant HIV-1 could regain infectivity through reversion mutations, they infected the HUT-CCR5 CD4⁺ T-cell line (herein HUT) with HIV-1 PR mutants pseudotyped either with WT HIV-1 Env or the vesicular stomatitis virus envelope glycoprotein (VSV-G). Both HIV-1 Env- and VSV-G-pseudotyped PR mutants, replicating in the T cell line, reverted to infectious viruses after 7-13 days post-infection. Furthermore, sequencing the PR showed that the P534 was indeed reverted to the original S534 in all but one clone.

Thus, they concluded that the aa 534 is critical for Env fusogenicity and that Env mutants can rapidly recover replication

competency in CD4+ T cells.

This work is interesting and clearly presented. However, some more experiments are necessary to strengthen the author's conclusions further.

1. The authors only showed pictures of HUT where some syncytia were detectable. It would be more informative to follow viral replication in this cell line by flow cytometry and monitor the appearance of Gag-positive cells.
2. Infectious viruses appeared around days 7 to 13: it could be expected that once the infectious virus is present in culture, it starts replicating efficiently in HUT cells. Is this the case?
3. Are the M1, M3, and M4 mutants able to bind to CD4? It is not clear why relative infectivity shown in figures 1D and 2C remains so low for the mutants even though they reverted to the S534. Is there any possible competition for CD4 binding between the "original" PR mutants lasting in the supernatant and the newly produced reverted viruses?
4. Authors should recover viral particles from HUT cells as they did, infect new HUT cells, and then test the infectivity of the viruses produced from this "second round" of HUT infection.
5. It would be important to test if these mutants in primary CD4+ T cells can revert as observed using the HUT T cell line.

Reviewer #2 (Comments for the Author):

The authors describe a well-conceived and conducted experimental evolution study to explore the revertants generated from mutation(s) in the polar region of HIV envelope glycoprotein gp41. The findings support these author's previous studies indicating the importance of, for example, serine at position 534 for HIV infection, and defines the Env residues critical for HIV-1 infection.

The authors do not speculate on the precise role of these polar residues in gp41 for HIV infection. I note with interest that recent studies have reported O-glycosylation of gp120 (Silver et al., 2020, Cell Reports 30, 1862-1869 February 11, 2020. <https://doi.org/10.1016/j.celrep.2020.01.056>). The key serine and threonine residues described in the current study may be targets for O-glycosylation, and this may be an optional discussion point that the authors may wish to consider addressing.

Minor typo:

Line 108 "collected at 3 to15 dpi to" space missing between "to" and "15"

Staff Comments:

Preparing Revision Guidelines

Please return the manuscript within 60 days; if you cannot complete the modification within this time period, please contact me. If you do not wish to modify the manuscript and prefer to submit it to another journal, please notify me of your decision immediately so that the manuscript may be formally withdrawn from consideration by Microbiology Spectrum.

Corresponding authors may join or renew ASM membership to obtain discounts on publication fees. Need to upgrade your

membership level? Please contact Customer Service at Service@asmusa.org.

Authors' point-by-point responses to reviewers' questions and comments

Lu *et al.* Reverted HIV-1 mutants in CD4+ T-cells reveal critical residues in the polar region of viral envelope glycoprotein.

Reviewer #1 comments and authors' responses

General comments: The paper presented by Wu and colleagues is a follow-up to a study previously conducted in the lab, which showed the critical role of the Env polar region (PR) of HIV-1 in controlling fusion between virus and cells and between cells. They previously have identified three mutants that they named M1, M3, and M4, all completely unable to fuse with target cells. However, these mutants, if pseudotyped with the HIV-1 WT Env protein, were fusogenic.

To examine whether M1, M3, and M4 PR mutant HIV-1 could regain infectivity through reversion mutations, they infected the HUT-CCR5 CD4+ T-cell line (herein HUT) with HIV-1 PR mutants pseudotyped either with WT HIV-1 Env or the vesicular stomatitis virus envelope glycoprotein (VSV-G). Both HIV-1 Env- and VSV-G-pseudotyped PR mutants, replicating in the T cell line, reverted to infectious viruses after 7-13 days post-infection. Furthermore, sequencing the PR showed that the P534 was indeed reverted to the original S534 in all but one clone.

Thus, they concluded that the aa 534 is critical for Env fusogenicity and that Env mutants can rapidly recover replication competency in CD4+ T cells.

This work is interesting and clearly presented. However, some more experiments are necessary to strengthen the author's conclusions further.

General responses: We thank reviewer #1 for the positive evaluation, constructive comments, and helpful suggestions. We would like to respond to the comments individually as follows.

Comment 1. The authors only showed pictures of HUT where some syncytia were detectable. It would be more informative to follow viral replication in this cell line by flow cytometry and monitor the appearance of Gag-positive cells.

Response 1: The images of HIV-1 induced syncytia in Hut/CCR5 cells indicated that HIV-1 Env- and VSV-G-pseudotyped PR mutants lead to efficient cell-cell fusion of Hut/CCR5 cells at 7-13 days post-infection (dpi). In contrast, replication-competent wild-type (WT) HIV-1 infected Hut/CCR5 cells showed strong syncytia at 3 dpi (Fig. 1C and Fig. 2B). To measure kinetics of WT and mutant virus infectivity during passaging in Hut/CCR5 cells, we collected supernatants of HIV-1-infected Hut/CCR5 cells every two days from 3 to 15 dpi to quantify relative HIV-1 infectivity using luciferase-expressing TZM-bl cells with equal viral input (Fig. 1D and Fig. 2C).

We agree that the images of syncytia are qualitative and appreciate the reviewer's suggestion to monitor productive infection in the Hut/CCR5 cells for Gag-positive cells by flow cytometry. We have previously attempted this approach. However, it is very challenging to use a flow cytometry-based assay to accurately measure the extremely low levels of viral replication occurring in Hut/CCR5 cells infected with HIV-1 Env- and VSV-G-pseudotyped PR mutants. Instead, we chose to monitor the infectivity of progeny virus derived from Hut/CCR5 cells by infection of TZM-bl reporter cells (Fig. 1D and Fig. 2C). TZM-bl cells express luciferase upon productive HIV-1 infection and they are highly sensitive to HIV-1 infection, which is much more robust and sensitive than intracellular immunostaining of Gag-positive cells and detection by flow cytometry.

Comment 2. Infectious viruses appeared around days 7 to 13: it could be expected that once the infectious virus is present in culture, it starts replicating efficiently in HUT cells. Is this the case?

Response 2: We thank reviewer for this important question and we agree that it is the case. We have added the discussion: “It is likely that, once an infectious HIV-1 was present in cultures, it started replicating efficiently in Hut/CCR5 cells.” (lines 101-102 in revised text).

In this experiment, supernatants of HIV-1-infected Hut/CCR5 cells were collected every two days from 3 to 15 dpi to quantify viral infectivity using TZM-bl cells with equal p24 input. To avoid potential confusion, we changed the x-axis label of Fig. 1D and Fig. 2C to “Days of HIV-1 collected from Hut/CCR5 cells”.

Comment 3. Are the M1, M3, and M4 mutants able to bind to CD4? It is not clear why relative infectivity shown in figures 1D and 2C remains so low for the mutants even though they reverted to the S534. Is there any possible competition for CD4 binding between the "original" PR mutants lasting in the supernatant and the newly produced reverted viruses?

Response 3: We appreciate this important question. Although we did not directly measure CD4 binding to the M1, M3, and M4 mutants, we would think that the binding may not be affected for two reasons: **(1)** HIV-1 Env binding to CD4 is dependent on gp120, but not gp41; **(2)** All three Env mutations in our study are in the PR of gp41, which would not directly affect the CD4-binding site in gp120.

Furthermore, our previous published data from virion-cell and cell-cell fusion assays clearly indicated that the mutant Env are defective or impaired in fusion (cited reference 10). Below are two key results supporting the conclusion. Our new results in this study are consistent with our published results, suggesting that the PR mutants mainly impair the fusogenicity of Env.

Fig. 3F (from Lu et al JVI 2019, cited reference 10). Virion-cell fusion was determined by flow cytometry-based BlaM-Vpr assays using TZM-bl cells (10 or 50 ng of p24 for HIV-1 with or without Env trans-supplementation, respectively). All experiments were performed with triplicate samples for panel F and repeated at least three times, and means +/- standard errors of the means are shown. Dunnett’s multiple-comparison test was used for statistical analysis. ***, P < 0.0001, for the comparison of the result with an individual mutant to that with WT HIV-1.

Fig. 4D (from Lu et al JVI 2019, cited reference 10). PR mutations abolish or significantly decrease Env-mediated cell-cell fusion. Transfected HEK293T cells were cocultured with TZM-bl cells for 24 h and then lysed for firefly luciferase activity measurement of Env-mediated cell-cell fusion. Average percentages of HIV-1 Env-mediated cell-cell fusion from three independent experiments are shown, with the value for the WT set as 100%. All experiments were performed with triplicate samples and repeated at least three times, and means +/-standard errors of the means are shown. Dunnett’s multiple-comparison test was used for statistical analysis. ***, P < 0.0001, for the comparison of the result with an individual mutant Env to that with WT Env.

Comment 4. Authors should recover viral particles from HUT cells as they did, infect new HUT cells, and then test the infectivity of the viruses produced from this "second round" of HUT infection.

Response 4: This is an interesting idea. We have passaged the viruses by adding fresh Hut/CCR5 cells in the original cultures (Fig. 1B and 2A). We then took the supernatants of the infected Hut/CCR5 cells and did the suggested assay using TZM-bl reporter cells because the cell line can be very sensitive for quantification of HIV-1 infectivity.

Comment 5. It would be important to test if these mutants in primary CD4+ T cells can revert as observed using the HUT T cell line.

Response 5: We fully agree with the reviewer regarding this important suggestion. In our grant applications to continue this study, we did propose using primary CD4+ T cells and HIV-1 clinical isolates in our future studies. Unfortunately, the proposed studies were not funded after initial submission and second-round revision. Additional experiments involving primary cells are currently beyond the scope of what we are capable of providing. We hope to obtain funding to perform more studies using primary CD4+ T cells in the future.

Reviewer #2 comments and authors' responses

General comments: The authors describe a well-conceived and conducted experimental evolution study to explore the revertants generated from mutation(s) in the polar region of HIV envelope glycoprotein gp41. The findings support these author's previous studies indicating the importance of, for example, serine at position 534 for HIV infection, and defines the Env residues critical for HIV-1 infection.

General responses: We thank the reviewer for the very supportive evaluation of our work and helpful suggestions. We responded to the reviewer's specific comments as follows:

Comment 1. The authors do not speculate on the precise role of these polar residues in gp41 for HIV infection. I note with interest that recent studies have reported O-glycosylation of gp120 (Silver et al., 2020, *Cell Reports* 30, 1862-1869 February 11, 2020. <https://doi.org/10.1016/j.celrep.2020.01.056>). The key serine and threonine residues described in the current study may be targets for o-glycosylation, and this may be an optional discussion point that the authors may wish to consider addressing.

Response 1. We thank the reviewer for the intriguing idea and helpful suggestion. The published study showed that a subset of patient-derived HIV-1 isolates contain O-linked carbohydrate on the variable 1 (V1) domain of Env gp120, but not on gp41 (Silver et al., *Cell Reports*, 2020). Although this is an important finding, we did not discuss this paper due to the lack of the evidence of O-linked glycosylation on the polar region of gp41.

To speculate on the precise role of these polar residues in gp41 for HIV-1 infection, we added discussions of our published structural analysis of the gp41 PR (Lu et al, *JVI* 2019, cited reference 10). Our results suggest that the PR of gp41, particularly the key residue S534, is structurally essential for maintaining the HIV-1 Env trimer, viral fusogenicity and infectivity (lines 140-144 in revised text).

Comment 2: Minor typo: Line 108 "collected at 3 to15 dpi to" space missing between "to" and "15"

Response 2: We appreciate the reviewer's careful evaluation of our manuscript. We apologize for the typo and we have corrected it.

November 16, 2021

Dr. Li Wu
The University of Iowa
Microbiology and Immunology
51 Newton Road
Iowa City 52242

Re: Spectrum01653-21R1 (Reverted HIV-1 mutants in CD4⁺ T-cells reveal critical residues in the polar region of viral envelope glycoprotein)

Dear Dr. Li Wu:

Your manuscript has been accepted, and I am forwarding it to the ASM Journals Department for publication. You will be notified when your proofs are ready to be viewed.

Sincerely,

Yongjun Sui
Editor, Microbiology Spectrum
